# The Forest Fire Dynamic Change Influencing Factors and the Impacts on Gross Primary Productivity in China

**Lili Feng** [1,2] and **Wenneng Zhou** [3,*]

1   Key Laboratory of Ecosystem Network Observation and Modeling, Institute of Geographic Sciences and Natural Resources Research, Chinese Academy of Sciences, Beijing 100101, China
2   National Ecosystem Science Data Center, Institute of Geographic Sciences and Natural Resources Research, Chinese Academy of Sciences, Beijing 100101, China
3   Guangdong Provincial Key Laboratory of Water Quality Improvement and Ecological Restoration for Watersheds, School of Ecology, Environment and Resources, Guangdong University of Technology, Guangzhou 510006, China
*   Correspondence: zhouwn@gdut.edu.cn

**Abstract:** Forest fire as a common disturbance has an important role in the terrestrial ecosystem carbon cycling. However, the causes and impacts of longtime burned areas on carbon cycling need further exploration. In this study, we exploit Thematic Mapper (TM) and Moderate Resolution Imaging Spectroradiometer (MODIS) data to develop a quick and efficient method for large-scale forest fire dynamic monitoring in China. Band 2, band 4, band 6, and band 7 of MOD09A1 were selected as the most sensitive bands for calculating the Normalized Difference Fire Index (NDFI) to effectively estimate fire burned area. The Convergent Cross Mapping (CCM) algorithm was used to analyze the causes of the forest fire. A trend analysis was used to explore the impacts of forest fire on Gross Primary Productivity (GPP). The results show that the burned area has an increased tendency from 2009 to 2018. Forest fire is greatly influenced by natural factors compared with human factors in China. But only 30% of the forest fire causes GPP loss. The loss is mainly concentrated in the northeast forest region. The results of this study have important theoretical significance for vegetation restoration of the burned area.

**Keywords:** forest fire; MOD09A1; NDFI; GPP; China

## 1. Introduction

Forest fire occurs globally on various scales every year, causing economic, social, ecological, and environmental damage [1]. It is caused by natural phenomena as well as anthropogenic activities. It results in severe damage to wildlife, fertile forest floors, timber, human property, and certain rare plant species [2]. As a result of the pervasive harmful effects due to forest fires, they have received increasing recognition in past recent years. Therefore, precise mapping and monitoring of the location and temporal distribution of wildfires are important. In addition, forest fire as one of the most significant natural disturbance processes would also modify the structure and the composition of the vegetation [3,4]. It is closely related to the carbon cycling and greenhouse gas emissions [5,6]. The forest fire can burn away dead or decaying vegetation, facilitating the growth of new trees and burned trees or the soil surface, releasing large amounts of carbon. Forest fire disturbances have an important impact on ecosystem stability and renewal, and succession of forest ecosystems. [7,8]. The effects of forest fires on ecosystems are not clear for a long time. Long-time burned area mapping is a critical factor to investigate the causes of the forest fire and the impacts of forest fires on Gross Primary Productivity (GPP). This research provides theoretical support for intelligent management and prediction of forest fires to achieve carbon neutrality in China.

Traditionally, a field survey was used to monitor forest fires. But this method is time-consuming and laborious and lacks spatial information. To overcome this problem, remote sensing is one of the most efficient and cost-effective techniques for fire detection and mapping. At present, there are so many forest fire remote sensing monitoring products that provide fire data sources, but these data have their limitations in terms of both spatial and temporal resolution. Fornacca et al. (2017) compared four common fire products such as MCD45A1, MCD64A1, MCD14ML, and Fire_CCI [9]. Each product has its limitations in terms of accuracy in different fire ranges. Therefore, the timely and accurate mapping of burned areas is essential for fire management and climate change. The remote sensing technique provides a labor-efficient method to quickly locate the burned area distribution and detect the impacts and causes of forest fires [10,11]. In the past decades, extensive studies have been carried out on the detection of burned areas by remote sensing [3,10,12]. Medium-resolution satellites such as the Landsat series are widely utilized to identify the burned area. However, they cannot accurately depict the burned area change because of the limitations of the temporal resolution. Combining information from different sensors can fill the gaps between high temporal resolution and medium spatial resolution [13]. High temporal resolution satellite images such as MODIS and AVHRR greatly improve the capability to detect the long-time burned area [14–16]. Accurate forest fire distribution provides the basis for subsequent studies. It is significant to acquire the long-time fire distribution for studying its causes and effects.

Forest fire occurs every year in China, especially during the dry season. These fires are due to various factors such as dry weather, flammable materials, and human action [13]. The natural factor is the main determinant of natural forest fire [17]. Climate change affects the occurrences and dynamics of fire by changing meteorological factors such as air temperature, precipitation, and humidity [18]. The two possible main factors associated with climatic change are temperature and precipitation [19]. Pausas et al. (2004) found that the temperature and precipitation were significantly correlated with the burned areas [20]. Westerling et al. (2006) found that the forest fire was caused by reduced winter precipitation together with earlier spring snowmelt [21]. Lehmann et al. (2014) found that burned areas in Australia had a strong correlation with the average precipitation [22]. Wu et al. (2014) found that climate was the primary factor compared with human activity [23]. To sum up the above, the mechanisms and interactions leading to forest fires in China are poorly understood. The causes of forest fires need further exploration.

Forest ecosystem carbon cycling is an important component of the entire terrestrial ecosystem. The most significant impact of a forest fire can be seen in vegetation. Forest fire disturbances have a significant impact on ecosystem stability and sustainability [7,24]. On the one hand, plants usually die instantly due to considerably severe forest fires. Frequent and high-intensity fires will lead to permanent changes in ecosystems and their components [25]. On the other hand, forest fires have been an important mechanism for generating ecological succession by acting as an environmental filter, selecting species, and shaping ecosystem communities [26]. Forest fire contributes significantly to climate change, consuming and transferring carbon to the atmosphere [27]. However, carbon change is usually ignored after a forest fire. Observational data suggest that vegetation growth and soil carbon content gradually recover over the following years [28]. If ecosystems can be restored to their pre-disturbance state, the terrestrial ecosystem carbon cycling remains in long-term dynamic equilibrium [29]. The effects of forest fire disturbance on ecosystems are not clear for a long time.

The study proposes a new method to map burned areas using multisensor remote sensing data by taking advantage of the high temporal resolution of Moderate-Resolution Imaging Spectroradiometer (MODIS) and the medium spatial resolution of Thematic Mapper (TM). In this study, a new spectral index called Normalized Difference Fire Index (NDFI) was derived from MODIS surface reflectance data to timely and accurately acquire the forest fire distribution and its dynamic change. The objectives of this study are as

follows: (1) to accurately obtain the spatiotemporal variations of fire distribution by NDFI; (2) to identify the causes of forest fires; and (3) to explore the impacts of forest fire on GPP.

## 2. Materials and Methods

### 2.1. Data and Data Processing

MOD09A1 data were downloaded from the National Aeronautics and Space Administration (NASA) (http://earthdata.nasa.gov/, accessed on 30 December 2020). It provides 8-days composite with 500 m spatial resolution data.

TM data with a spatial resolution of 30 m and a return cycle of 16 d were downloaded from the Institute of Remote Sensing and Digital Earth (http://ids.ceode.ac.cn/, accessed on 30 December 2020) and used to validate the forest fire distribution.

Meteorological data of temperature and precipitation datasets with 1 km spatial resolution were downloaded from the National Ecosystem Science Data Center (NESDC). The dataset is interpolated by the data from the National Meteorological Information Center (NMIC) of the China Meteorological Administration and the Daily Global Historical Climatology Network-Daily (GHCN-D).

Gross Primary Productivity (GPP) and nighttime-light data from 2009 to 2018 were downloaded from National Tibetan Plateau Data Center. The Advanced Very High-Resolution Radiometer (AVHRR) data of remote sensing and hundreds of flux stations around the world were used to generate the global high-resolution long-time series GPP. The unit of this data is $gcm^{-2}$ with a spatial resolution of 0.05 degrees. The nighttime-light data were produced by the convolutional Long Short-Term Memory network method.

### 2.2. Burned Area Monitoring Methods

Remote sensing is a more appropriate approach for large-scale and long-time studies. The monitoring method was developed by Feng. The methodology used to estimate the forest fire distribution from 2009 to 2018 was based on the approach proposed by Feng et al. (2016) [30]. This method is beneficial to the dynamic monitoring of long-time series over large areas. The most sensitive bands were used to construct the NDFI to effectively estimate the burned area distribution in this study [30]. The main processes include (1) Band 2, Band 4, Band 6, and Band 7 of MOD09A1 were selected as the most sensitive bands to forest fire; (2) the sensitive bands were selected to calculate NDFI for monitoring the burned area; (3) Convergent Cross Mapping (CCM) algorithm was used to analyze the causes of forest fires; and (4) trend analysis was used to explore impacts of forest fire on GPP.

In order to effectively monitor burned area distribution, TM2, TM4, TM5, and TM7 of TM or MODIS2, MODIS4, MODIS6, MODIS7 of MODIS were selected to calculate NDFI using the following equation:

$$NDFI = \frac{|TM7 - TM5|}{TM4 + TM2} \tag{1}$$

$$NDFI = \frac{|MODIS7 - MODIS6|}{MODIS4 + MODIS2} \tag{2}$$

where *NDFI* is the Normalized Difference Fire Index; *TM7* is band 7 of *TM*; *TM5* is band 5 of *TM*; *TM4* is band 4 of *TM*; *TM2* is band 2 of *TM*. *MODIS7* is band 7 of MODIS; *MODIS6* is band 6 of MODIS; *MODIS4* is band 4 of MODIS; *MODIS2* is band 2 of MODIS.

### 2.3. Causes and Impacts of Forest Fire Analysis Methods

CCM algorithm proposed by Sugihara et al. was initially applied to detect the causality of variables in complex ecosystems. It is a powerful new methodological approach that can help distinguish causality from spurious correlation in time series from dynamic systems. The technique is based on the idea that causation can be established if states of the causal variable can be recovered from the time series of the affected variable [31]. It uses Takens'

idea to detect if two variables belong to the same dynamical system. It is designed for causal discovery between coupled time series for which Granger's method for detecting causality is shown to be unreliable. CCM is based on an algorithm that compares the ability of lagged components of one process to estimate the dynamics of another. In ecology, these processes might represent time series observations of environmental data, such as temperature, or of species data, such as population abundance [32]. In this study, we did not replace causality with correlation. CCM algorithm was used to analyze the causes of the forest fire. The forest fire, temperature, precipitation, and nighttime-light time series data were used to derive the causality.

A slope map was calculated at pixel scales to evaluate the spatial change of burned area and GPP using the following equation:

$$slope = \frac{n \sum_{j=1}^{n} jy - \sum_{j=1}^{n} j \sum_{j=1}^{n} y}{\sum_{j=1}^{n} j^2 - \left(\sum_{j=1}^{n} j\right)^2} \tag{3}$$

where $n$ is the number of years, here $n = 10$; $y$ is the burned area (GPP) in the $j$th year; the *slope* is the fitted slope of $n$ years. The slope map indicates the variation in trend and range.

## 3. Results

### 3.1. Sensitive Bands Selection of Forest Fire

In this study, the huge forest fire that originated in China's Daxinganling Mountains in 1987 was selected as the reference area which obviously reduces the influence of mixed pixel. Twenty random points were generated in the burned area (Figure 1). The DN values (data value range: 0~65,535) of these points in different bands of TM in different seasons are shown in Tables 1–3. During these three months, forest changes can be very pronounced in the event of a fire. Therefore, these three months are the basic months for the occurrence of fires. The total correlation index(r) value of all types between band 5 and band 7 is the lowest; band 5 and band 7 exhibit a small disparity in their spectral responses to different land covers. The total correlation index(r) value of all types between band 2 and band 4 is the largest; band 2 and band 4 exhibit a large disparity in their spectral responses to different land covers (Table 4). Therefore, these four bands are used to derive the Normalized Difference Fire Index (NDFI) in this study [33]. A lower NDFI value indicated a greater possibility of fire distribution.

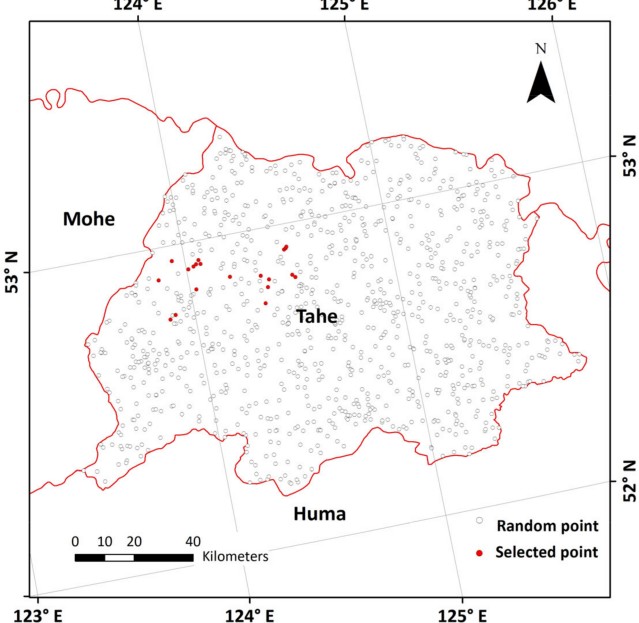

**Figure 1.** Twenty selected random points in the burned area.

**Table 1.** Twenty selected random points values of TM in April.

| Random Point | Band 1 | Band 2 | Band 3 | Band 4 | Band 5 | Band 7 |
|---|---|---|---|---|---|---|
| 1 | 9523 | 10,286 | 10,777 | 13,353 | 15,720 | 12,999 |
| 2 | 9273 | 9832 | 10,101 | 13,024 | 14,135 | 11,495 |
| 3 | 9371 | 9797 | 9937 | 12,098 | 13,297 | 10,896 |
| 4 | 9315 | 9653 | 9812 | 12,420 | 13,729 | 11,055 |
| 5 | 9107 | 9449 | 9924 | 12,278 | 14,689 | 12,111 |
| 6 | 9530 | 9806 | 10,217 | 12,858 | 13,811 | 11,191 |
| 7 | 9506 | 9790 | 10,209 | 12,862 | 14,642 | 11,785 |
| 8 | 9139 | 9579 | 9758 | 12,133 | 13,045 | 10,932 |
| 9 | 9056 | 9626 | 9932 | 12,123 | 14,164 | 11,810 |
| 10 | 9542 | 10,130 | 10,634 | 14,231 | 15,157 | 12,229 |
| 11 | 9281 | 9948 | 10,490 | 14,294 | 14,701 | 11,821 |
| 12 | 9772 | 10,295 | 10,500 | 12,253 | 11,543 | 10,157 |
| 13 | 9462 | 9597 | 10,049 | 12,715 | 14,345 | 11,793 |
| 14 | 9134 | 9630 | 9794 | 12,574 | 13,009 | 10,909 |
| 15 | 9225 | 9751 | 10,464 | 13,177 | 16,109 | 12,983 |
| 16 | 9813 | 9998 | 10,236 | 11,943 | 11,844 | 10,152 |
| 17 | 9676 | 10,004 | 9966 | 12,698 | 12,870 | 10,739 |
| 18 | 10,016 | 10,465 | 11,052 | 13,461 | 15,553 | 12,366 |
| 19 | 9375 | 9845 | 10,106 | 12,861 | 12,574 | 10,748 |
| 20 | 9293 | 9682 | 9967 | 13,162 | 13,088 | 10,895 |

**Table 2.** Twenty selected random points values of TM in May.

| Random Point | Band 1 | Band 2 | Band 3 | Band 4 | Band 5 | Band 7 |
|---|---|---|---|---|---|---|
| 1 | 9120 | 9162 | 9267 | 9566 | 12,734 | 13,833 |
| 2 | 7500 | 7953 | 8252 | 8762 | 11,776 | 13,059 |
| 3 | 8710 | 8912 | 8853 | 9353 | 11,454 | 12,371 |
| 4 | 7546 | 7988 | 8120 | 8777 | 10,736 | 12,030 |
| 5 | 6882 | 7460 | 7844 | 8366 | 11,578 | 12,349 |
| 6 | 8623 | 8746 | 8845 | 9049 | 10,583 | 11,413 |
| 7 | 8615 | 8897 | 8840 | 9196 | 11,455 | 12,237 |
| 8 | 7082 | 7577 | 7943 | 8679 | 11,062 | 12,219 |
| 9 | 6759 | 7437 | 7826 | 8530 | 10,744 | 11,471 |
| 10 | 8594 | 8808 | 8917 | 9541 | 12,381 | 13,710 |
| 11 | 7890 | 8342 | 8555 | 9183 | 11,761 | 13,455 |
| 12 | 8413 | 8684 | 8668 | 9091 | 10,794 | 11,323 |
| 13 | 8293 | 8394 | 8422 | 8803 | 10,787 | 11,727 |
| 14 | 8612 | 8794 | 8615 | 9224 | 11,458 | 12,647 |
| 15 | 8172 | 8632 | 8630 | 9068 | 11,781 | 13,281 |
| 16 | 8707 | 8851 | 8917 | 9093 | 10,578 | 11,123 |
| 17 | 8717 | 8905 | 8840 | 9194 | 10,972 | 11,142 |
| 18 | 8440 | 8761 | 8849 | 9498 | 12,521 | 12,650 |
| 19 | 8541 | 8943 | 8734 | 9432 | 10,866 | 11,653 |
| 20 | 8440 | 8775 | 8978 | 9134 | 11,250 | 12,199 |

**Table 3.** Twenty selected random points values of TM in June.

| Random Point | Band 1 | Band 2 | Band 3 | Band 4 | Band 5 | Band 7 |
|---|---|---|---|---|---|---|
| 1 | 8297 | 8494 | 8370 | 9084 | 12,537 | 13,511 |
| 2 | 8275 | 8643 | 8776 | 9545 | 13,318 | 13,510 |
| 3 | 8284 | 8348 | 8243 | 8864 | 10,654 | 10,566 |
| 4 | 8366 | 8654 | 8785 | 9395 | 13,509 | 14,057 |
| 5 | 8464 | 8829 | 9068 | 10,166 | 14,288 | 14,329 |
| 6 | 8491 | 8648 | 8896 | 9606 | 12,488 | 13,162 |
| 7 | 8418 | 8800 | 8632 | 9008 | 10,751 | 11,524 |

**Table 3.** *Cont.*

| Random Point | Band 1 | Band 2 | Band 3 | Band 4 | Band 5 | Band 7 |
|:---:|:---:|:---:|:---:|:---:|:---:|:---:|
| 8 | 8477 | 8512 | 8795 | 9707 | 13,315 | 13,919 |
| 9 | 8323 | 8676 | 8800 | 9557 | 13,604 | 14,465 |
| 10 | 8599 | 8765 | 8732 | 9437 | 12,791 | 13,455 |
| 11 | 8807 | 9059 | 9386 | 10,340 | 14,468 | 15,411 |
| 12 | 8643 | 8644 | 8759 | 9302 | 11,915 | 12,622 |
| 13 | 8850 | 9005 | 9196 | 10,216 | 15,731 | 16,260 |
| 14 | 8398 | 8578 | 8712 | 9279 | 12,027 | 12,510 |
| 15 | 8738 | 8876 | 9471 | 10,809 | 14,190 | 12,859 |
| 16 | 8493 | 8734 | 8839 | 9723 | 12,312 | 12,368 |
| 17 | 8416 | 8735 | 8707 | 9423 | 12,408 | 13,052 |
| 18 | 8550 | 8568 | 8429 | 9114 | 11,535 | 12,782 |
| 19 | 8729 | 8810 | 8901 | 10,207 | 14,624 | 14,952 |
| 20 | 8635 | 8797 | 9024 | 9904 | 13,662 | 14,272 |

**Table 4.** Correlation coefficient (r) value between different bands.

|  | Band 1 | Band 2 | Band 3 | Band 4 | Band 5 | Band 7 |
|:---:|:---:|:---:|:---:|:---:|:---:|:---:|
| Band 1 | — — | 0.993081 | 0.991975 | 0.992561 | 0.908178 | −0.63607 |
| Band 2 | 0.993081 | — — | 0.999959 | 0.999991 | 0.85274 | −0.72228 |
| Band 3 | 0.991975 | 0.999959 | — — | 0.999989 | 0.847965 | −0.72853 |
| Band 4 | 0.992561 | 0.999991 | 0.999989 | — — | 0.850461 | −0.72528 |
| Band 5 | 0.908178 | 0.85274 | 0.847965 | 0.850461 | — — | −0.25467 |
| Band 7 | −0.63607 | −0.72228 | −0.72853 | −0.72528 | −0.25467 | — — |

### *3.2. NDFI Validity and Burned Area Recognition in China*

Further, the huge forest fire that originated in China's Daxinganling Mountains in 1987 was used to validate the validity of the NDFI. Burned area in China's Daxinganling Mountains is shown in Figure 2. TM2, TM4, TM5, and TM7 of TM are used to calculate the NDFI image. A lower value indicated a greater possibility of burned area. This result is consistent with the actual situation. The MODIS data were used to calculate the NDFI for estimating the burned area at a large scale. The possible burned area distribution in China is shown in Figure 3. A lower value indicated a greater possibility of forest fires. The northeast forest fire in 2010 is used to validate the burned area result obtained from MODIS data. The validation result (Figure 4) shows that the result is consistent with the actual forest fire distribution. Meanwhile, monitoring results are consistent with existing studies [34].

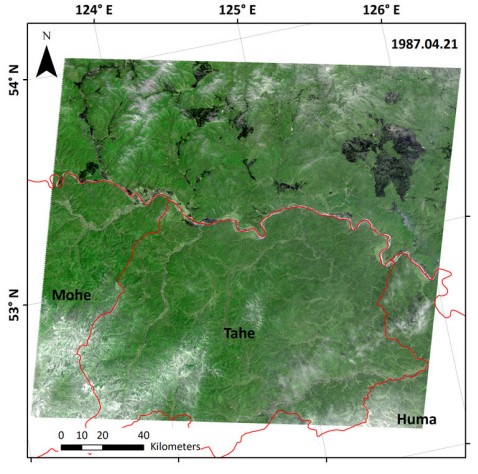 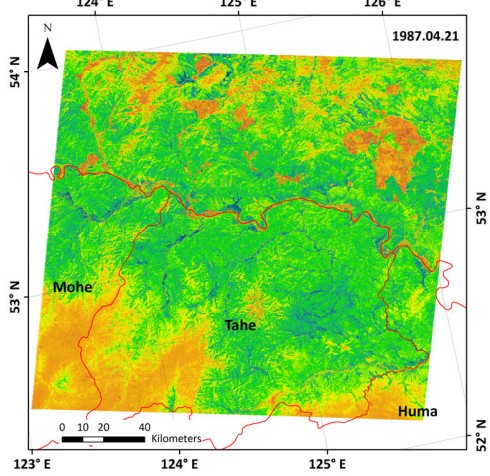

**Figure 2.** *Cont.*

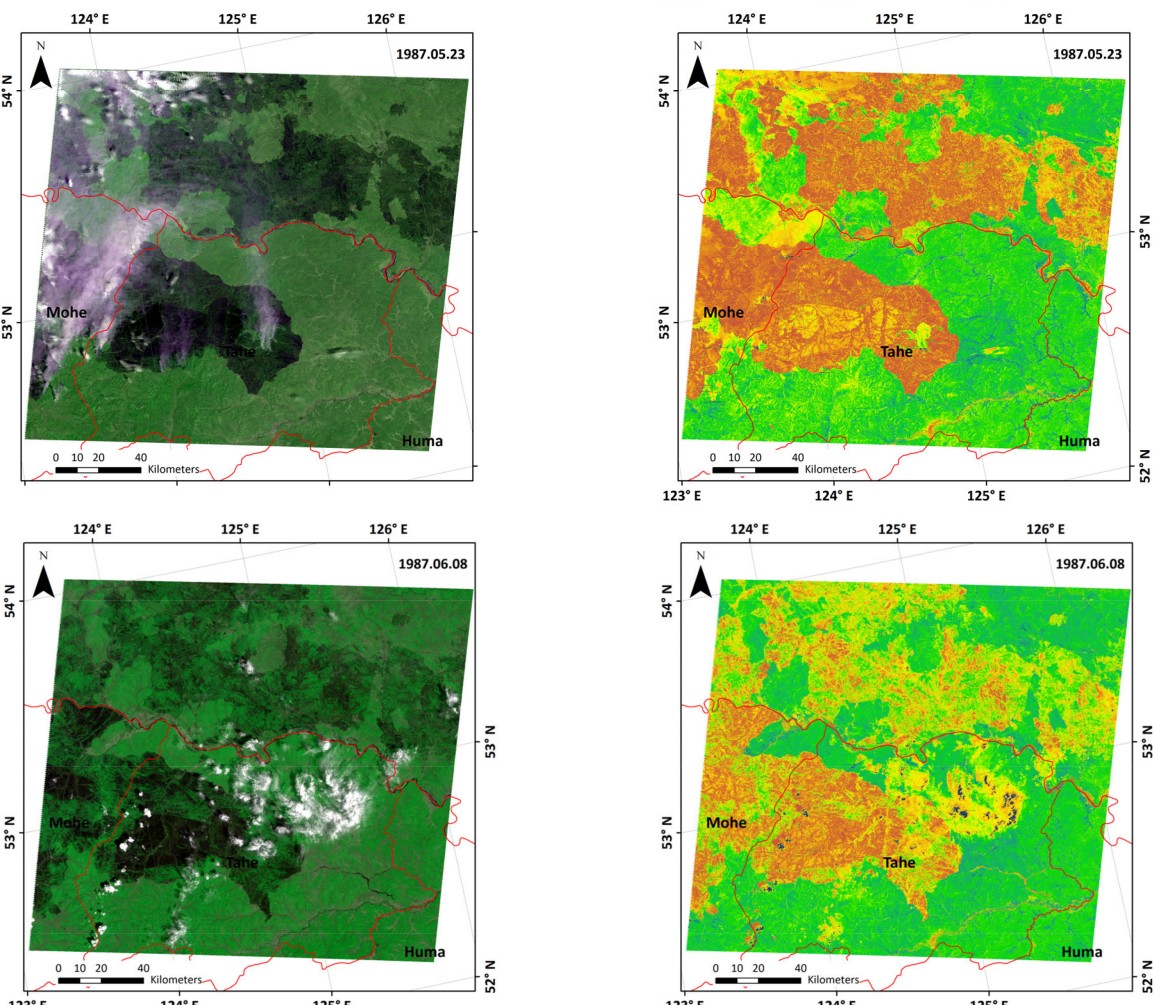

**Figure 2.** Raw TM (**left**) (3, 4, 2 bands) and NDFI image (**right**) in China's Daxinganling Mountains. TM2, TM4, TM5, and TM7 of TM is used to calculate the NDFI image. A lower value indicated a greater possibility of burned area.

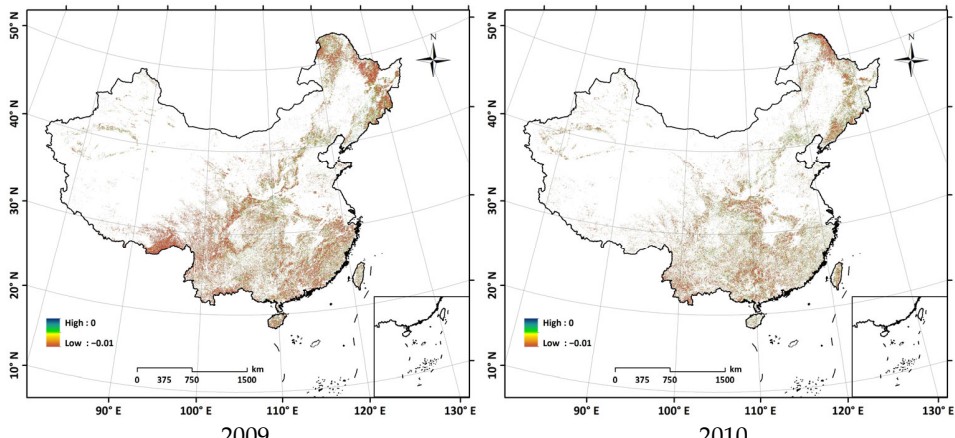

**Figure 3.** *Cont.*

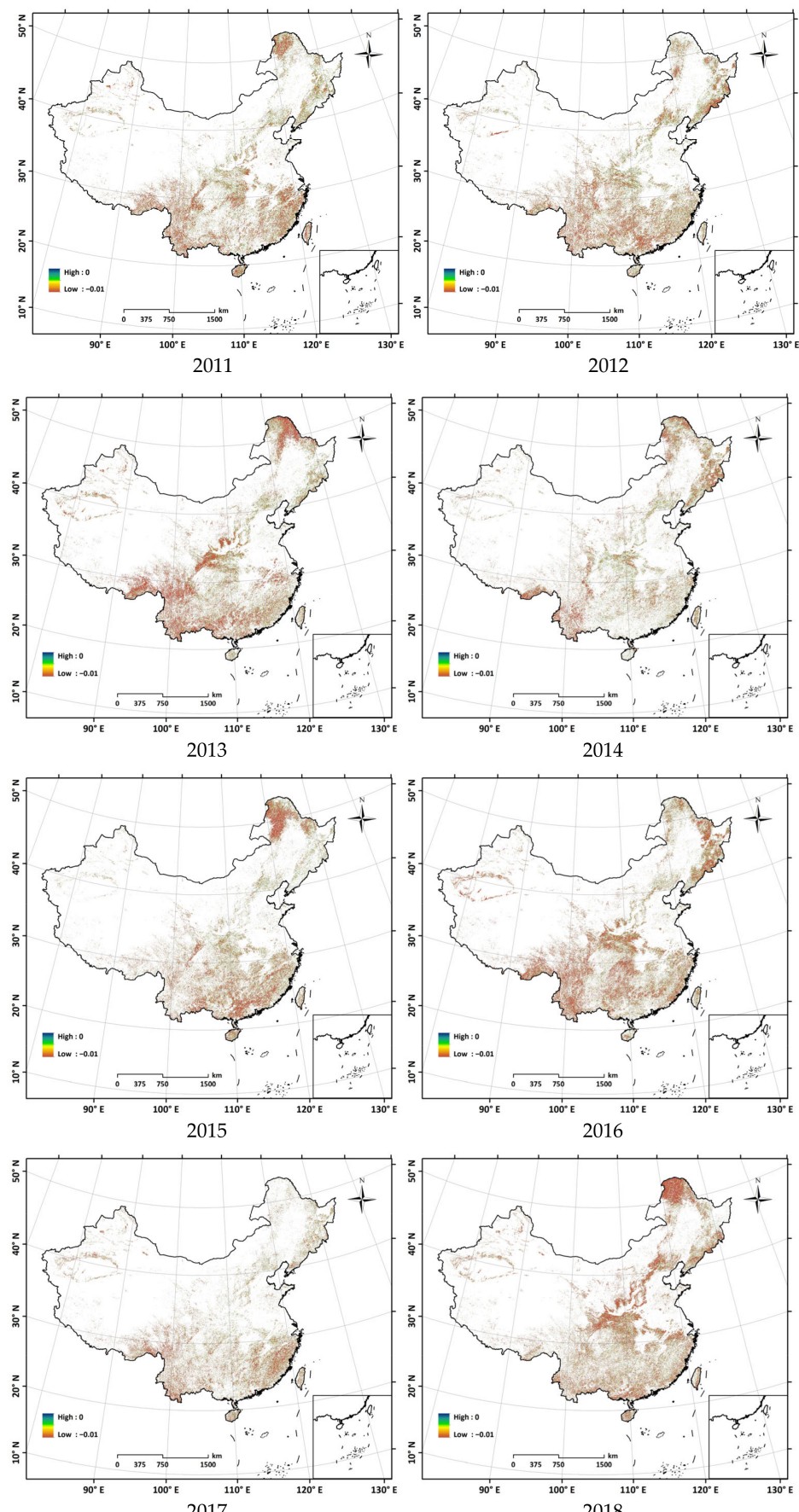

**Figure 3.** Possible fire distribution from 2009 to 2018 in China.

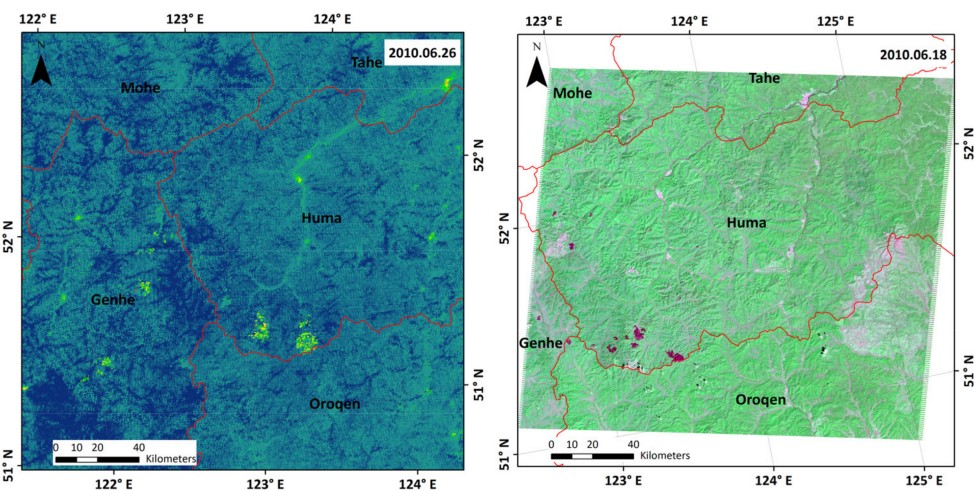

**Figure 4.** NDFI of MODIS and raw TM in 2010.

### 3.3. Causes of the Forest Fire

CCM of the fire with temperature, precipitation and nighttime-light time series data was shown in Figure 5. Pearson's correlation coefficient at the point of convergence for "B causes A" is greater than that for "A causes B", where A represents forest fire changes and B represents the temperature, precipitation, and nighttime-light changes, respectively (Figure 5a–c). That means all these three factors can drive the forest fire. The temperature and precipitation represent natural factors. The nighttime-light represents the human factor [35]. Results for this CCM analysis suggest that forest fire is significantly affected by temperature and precipitation (Figure 5a,b). However, there is no obvious forcing for the burned area and nighttime-light (Figure 5c). This result indicates that forest fire is greatly influenced by natural factors compared with human factors.

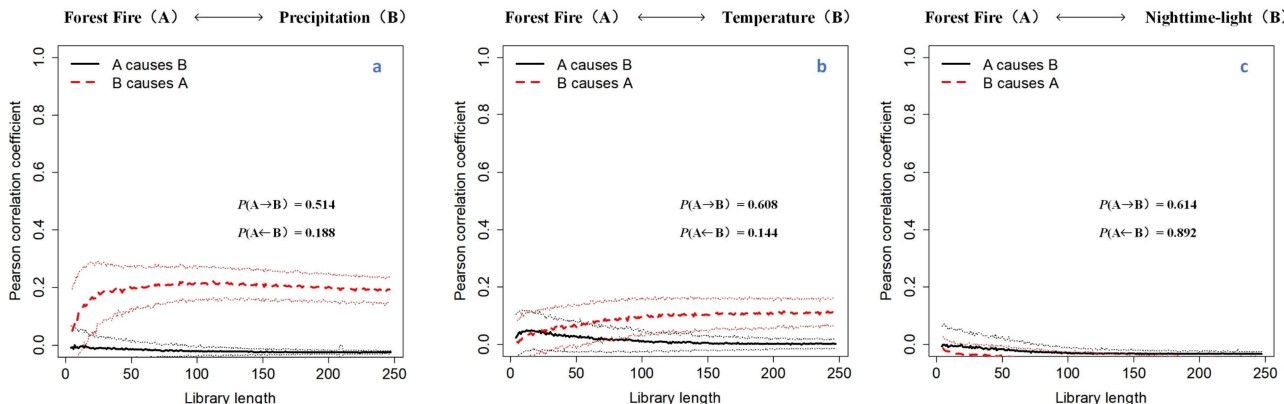

**Figure 5.** Convergent Cross Mapping of burned area with the possible influencing factors. (**a**) Pearson's correlation coefficient at the point of convergence for forest fire and precipitation changes; (**b**) Pearson's correlation coefficient at the point of convergence for forest fire and temperature changes; (**c**) Pearson's correlation coefficient at the point of convergence for forest fire and nighttime-light changes.

### 3.4. Effects of the Forest Fire on Ecosystem Carbon Cycle

The slope map of burned area and GPP from 2009 to 2018 is shown in Figure 6. Forest fire has the decreased trend. The GPP loss areas caused by forest fires are the regions where forest fire increased and the GPP decreased (Figure 7). The analysis result shows that about 30% of the forest fire causes GPP loss. The loss is mainly concentrated in the northeast forest region. However, about 70% of the forest fire have no impact on GPP.

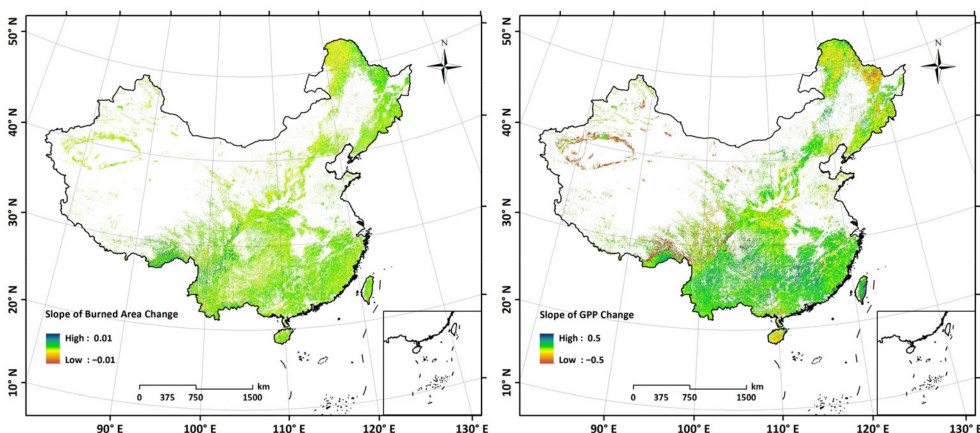

**Figure 6.** The slope map of burned area (**left**) and GPP (**right**) change from 2009 to 2018.

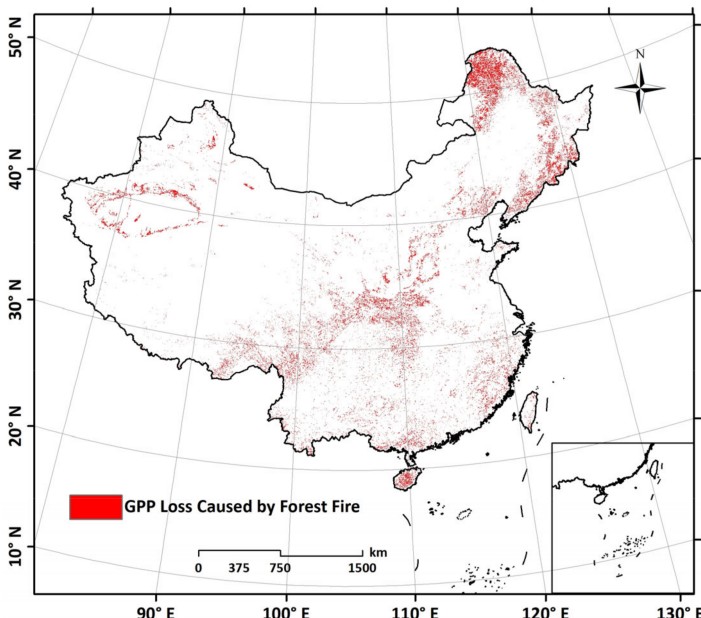

**Figure 7.** Forest fire causes the GPP loss area.

## 4. Discussion

### 4.1. Burned Area Distribution and Causes of Forest Fire

The burned area distribution is supported by previous studies. For example, Chen et al., (2017) have demonstrated that the burned area is mainly distributed in eastern China, especially in the forests of Heilongjiang Province in Northeast China [36]. Pang et al., denoted that the probability of forest fires in eastern China is higher than that in the western regions, and the probability of forest fires in the north and south is higher than that in Central China [37]. The results of these studies are consistent with our study. Meanwhile, forest fire is driven by various environmental factors [36]. Climatic change and anthropogenic factors are likely altering the fire distribution in most regions of China. The increasing forest fire in northeastern China may have resulted from an increased temperature and decreased precipitation and humidity [38]. In addition, longitude and latitude had the greatest influence on the occurrence of forest fires. This result is due to the uneven distribution of forest resources and regional differences in forest resources in China. The higher the vegetation cover the more likely they are to cause problems related to forest fires [37]. The Intergovernmental Panel on Climate Change states that "climate variability is often the dominant factor affecting large wildfires" [39]. Archibald et al., (2010) denoted that climate is a dominant control on fire activity, regulating vegetation productivity and fuel

moisture [40]. Precipitation suppresses the forest fire activity and promotes flammable material accumulation. Additionally, the temperature provides burning conditions for the flammable materials. Human activity is now the primary source of ignition in tropical forests, savannas, and agricultural regions [41,42]. In our study, the forest fire primarily located in Northeast China is mainly caused by natural factors. This result is consistent with some previous studies. However, we only discuss the temperature and precipitation as the natural factor and the nighttime-light as the human action. More factors and quantitative analysis methods should be discussed. Meanwhile, different data sources will bring a lot of uncertainty to the results. Therefore, multi-source data on fire range identification and GPP impact factor identification process will be considered in future work.

### 4.2. Practical Implications of This Study

The forest fire as a major environmental and social issue has attracted widespread attention. It is significant to propose a simple and effective method to estimate the longtime burned area on a large scale. Many fire products can be obtained from the website. However, accurate and longtime burned area distributions are limited. It is necessary to acquire the longtime burned area distribution to analyze the causes of the forest fire and the impacts of forest fire on GPP. In fact, there are many factors affecting GPP, which can be divided into natural and human factors [43]. On the one hand, forest fire as an important disaster will bring a certain loss to GPP. On the other hand, fire as a common and natural disturbance in forest regions plays a critical role in determining the structure and composition of vegetation [44,45]. Many plants have acquired the adaptive ability to regenerate after a forest fire to keep dynamic disequilibrium [29]. Fires have been found to be important in maintaining vegetation diversity, structure, and functions in fire prone ecosystems such as savannas. Forest fires do not necessarily bring species diversity changes [46]. A recent study showed that savannahs store carbon despite frequent fires [47]. So not all forest fires have a negative impact on ecosystems. In our study, during the revegetation for many years, 70% of the forest fire has no impact on GPP. The result of this study is useful to improve the ecosystem model [48] and forest management [49]. It is significant for promoting carbon neutrality in China. However, we only analyzed the impacts of forest fires on GPP at a national scale ignoring spatial heterogeneity and influence factor differences in different regions. That is the limitation of this study. We will consider these differences in future work.

### 5. Conclusions

The proposed NDFI is able to obtain the forest fire distribution on a large scale and assess the variability simply and effectively. The forest fire change has an increased tendency from 2009 to 2018. These fires are significantly affected by temperature and precipitation compared with nighttime-light. It indicates that forest fire is greatly influenced by natural factors. But only 30% of the forest fire causes GPP loss. The loss is mainly concentrated in the northeast forest region. This study provides an effective way to understand the burned area dynamic changes, the causes of the forest fire and the impacts of forest fire on GPP. However, there are still many challenges and uncertainty to estimating forest fire distribution for long time series observation on large scale using remote sensing data. Multi-source data on fire range identification should be considered and more factors and quantitative analysis methods should be considered in further studies.

**Author Contributions:** L.F. and W.Z. conceived and designed research. L.F. performed the experiments, analyzed the data, and wrote the manuscript. L.F. and W.Z. revised the manuscript. All authors have read and agreed to the published version of the manuscript.

**Funding:** This work was supported by the National Natural Science Foundation of China (42141005), National Key Research and Development Program of China (2021YFF0703900).

**Data Availability Statement:** All data can be accessed through the provided website and references.

**Acknowledgments:** We thank the staff of relevant departments for their dedication in observation and data processing. We also thank anonymous reviewers and partners of Weihua Liu, Mengyu Zhang, Yonghong Zhang, Tianxiang Wang, Yan Lv, Qingqing Chang, and Qian Xu for their good advice and their help.

**Conflicts of Interest:** The authors declare no conflict of interest.

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
