# Peer review of "The Forest Fire Dynamic Change Influencing Factors and the Impacts on Gross Primary Productivity in China"

_remotesensing, doi:10.3390/rs15051364_

Round 1

Reviewer 1 Report (Previous Reviewer 1)

A forest fire has been one of the premier significant issues. It is the most common hazard that brings harm to forests, the environment, wildlife and economics. Further, forest fire is an important part of forest carbon cycling. This research is important to improve the carbon models and help to analyze the effect of fire on carbon cycle. After revision, the article reached my expectation. Even though I have expressed lots of concerns in the last results, I still believe the study has a lot of potentials. Doing novel things is always difficult. I would encourage the authors to keep refining their study, and I sincerely hope My previous opinions are useful to improve their manuscript.

Author Response

Thank you very much for your comments. We have revised our paper as far as possible according to your helpful advice. We have listed relevant reference to compare with our study. Thank you very much for all your advice.

Reviewer 2 Report (Previous Reviewer 2)

Dear Authos,

I appreciate that you resumbitted the manuscript and, most of all, that you took into consideration my first remarks.

As I mentioned in my first review, your paper is interesting for your region and well written. Also, I have to mention that the new title is way better.

Please pay attention that you have so missing and additional spaces in the text. 

The figures are all very blury. Can you improve them?

Thank you and wish you good luck in your research activity!

Author Response

Reviewer #2:

Comment: I appreciate that you resumbitted the manuscript and, most of all, that you took into consideration my first remarks. As I mentioned in my first review, your paper is interesting for your region and well written. Also, I have to mention that the new title is way better.

Response: Thank you very much for your encouragement. We have carefully revised the manuscript according to your suggestions, and hope that you are satisfied with the revision.

Q1: Please pay attention that you have so missing and additional spaces in the text. The figures are all very blury. Can you improve them? Thank you and wish you good luck in your research activity!

A1: It is so helpful for us. Thank you very much for your advice. We have revised relevant parts in the revised manuscript as you suggested. We have also revised the blurred figures. At last, thanks very much for all your constructive and helpful comments and suggestions throughout the review process.

Reviewer 3 Report (Previous Reviewer 3)

This would be a very meaningful work, as I said to the first edition last time. Unfortunately, some of the issues mentioned last time are still not well explained, even I have not seen meaningful improvement in the revised version. The remaining problems include

(1) The relationship between NDFI and burned area needs to be explained in the method. How to recognize burned area by NDFI?

(2) What are the values, units and meanings listed in Table 1, 2 and 3?

(3)CCM method needs to be explained.

(4)The results are difficult to support the discussion and conclusions

(5) The pictures are not clear.

Author Response

Reviewer #3:

Comment: This would be a very meaningful work, as I said to the first edition last time. Unfortunately, some of the issues mentioned last time are still not well explained, even I have not seen meaningful improvement in the revised version.

Response: Thank you very much for your comments. We have revised the first edition as far as possible and we are so sorry for the imperfect modification. So we will add further explanations in this edition as your advice. Thank you very much for all your advice.

Q1: The relationship between NDFI and burned area needs to be explained in the method. How to recognize burned area by NDFI?

A1: Thanks for your advices. Indeed, the explanation is not clear. So we have added more descriptions to the revised manuscript.

The total correlation index(r) value of all types between band5 and band7 is the lowest, band5 and band7 exhibit a small disparity in their spectral responses to different land covers. The total correlation index(r) value of all types between band2 and band4 is the largest, band2 and band4 exhibit a large disparity in their spectral responses to different land covers (Table 4). So, these four bands are used to derive the Normalized Difference Fire Index (NDFI) in this study. A lower NDFI value indicated a greater possibility of fire distribution.

Q2: What are the values, units and meanings listed in Table 1, 2 and 3?

A2: Thanks for your questions. Indeed, we did not describe the Table clearly. The value is the DN value (data value range: 0~ 65535), these values have no units. We have added relevant description in the revised manuscript.

Q3: CCM method needs to be explained.

A3: It is so helpful for us. Thank you very much for your advice to make our manuscript more accurate. As you suggested, we have added relevant description in the revised manuscript. Please see the method part.

It uses Takens' idea to detect if two variables belong to the same dynamical system. It is designed for causal discovery between coupled time series for which Granger's method for detecting causality is shown to be unreliable. CCM is based on an algorithm that compares the ability of lagged components of one process to estimate the dynamics of another. In ecology, these processes might represent time series observations of environmental data, such as temperature, or of species data, such as population abundance. In this study, we did not replace causality with correlation. CCM algorithm was used to analyze the causes of the forest fire. Forest fire, temperature, precipitation and nighttime-light time series data were used to derive the causality.

Q4: The results are difficult to support the discussion and conclusions.

A4: Thank you very much for your good suggestion. We have revised the related parts in the revised manuscript as your advice, please see the Discussion and Conclusion part.

Q5: The pictures are not clear.

A5: Thank you very much for your advice. We have improved the blurred figures.

Reviewer 4 Report (Previous Reviewer 4)

This study exploit Thematic Mapper (TM) and Moderate Resolution Imaging Spectroradiometer (MODIS) data to develop a quick and efficient method for large-scale forest fire dynamic monitoring in China and explore the impacts of forest fire on Gross Primary Productivity (GPP). In general, the idea is clear, the structure is reasonable, the article has certain practical significance. Some suggestions:

1.  The second paragraph of the introduction lacks an overview of current fire identification methods. What are the methods? What are the advantages and disadvantages? What is the innovation of the method used in this study.

2.  Fire identification requires high data accuracy. In this study, the impact of fire on regional GPP was explored, and whether the impact of multi-source data on fire range identification and GPP impact factor identification process was considered.

3.  From the result, the possibility of fire in Northeast China is high. Why? Did the author consider the reason?

4.  In the discussion part, although the author discussed the impact of fire on GPP, there are still some deficiencies. In fact, there are many factors affecting GPP, which can be divided into natural and human factors. Many scholars try to separate the contribution rate of natural factors and human factors to GPP, such as What drives the vegetation dynamics in the Hengduan Mountain region, southwest China: Climate change or human activity? DOI: 10.1016/j.ecolind.2019.106013. Interestingly, fire can be either a natural factor or an artificial factor. From this point of view, can we discuss it further?

Author Response

Reviewer #4:

Comment: This study exploit Thematic Mapper (TM) and Moderate Resolution Imaging Spectroradiometer (MODIS) data to develop a quick and efficient method for large-scale forest fire dynamic monitoring in China and explore the impacts of forest fire on Gross Primary Productivity (GPP). In general, the idea is clear, the structure is reasonable, the article has certain practical significance.

Response: Thank you very much for your comments and kind encouragement on our manuscript. We will revise our paper as far as possible according to your helpful advice. Thank you very much for all your advice.

Q1: The second paragraph of the introduction lacks an overview of current fire identification methods. What are the methods? What are the advantages and disadvantages? What is the innovation of the method used in this study.

A1: It is so helpful for us. Thank you very much for your comments. We will revise our paper as far as possible according to your helpful advice. Thank you very much for all your advice. Meanwhile, I urgently need to clarify the importance and innovation of our research. First, forest fire has been one of the premier significant issues. It is the most common hazard that brings harm to forests, environment, wildlife and economics. Further, the forest fire is an important part of the ecosystem carbon cycling. Until now, we are still not sure about the impact of forest fires on carbon sink. In our research group, we develop several models to estimate the carbon sink that do not take into account the effects of fire. So we should acquire accurate forest fire distribution and consider the forest fire effects on the carbon model. This research is important to improve our carbon models and help to achieve carbon neutrality in China. That is the background of our research.

Meanwhile, remote sensing is a more appropriate approach for large scale and longtime studies. The monitoring method was developed by Feng. And the method was used in various applications such as corn and aeolian desertification distribution monitoring (Zhang & Feng et al., 2014; Feng et al., 2016). The method used in this research improves monitoring accuracy. Otherwise, the Convergent Cross Mapping (CCM) method is a very effective method to research causality. CCM is a powerful new methodological approach to detect causality between time series (Sugihara et al. 2012). It uses Takens' idea to detect if two variables belong to the same dynamical system. It is designed for causal discovery between coupled time series for which Granger's method for detecting causality is shown to be unreliable. In our study, we did not replace causality with correlation. It is reasonable to study the causes of forest fires using CCM.

Zhang J, Feng L, Yao F. Improved maize cultivated area estimation over a large scale combining MODIS–EVI time series data and crop phenological information[J]. ISPRS Journal of Photogrammetry and Remote Sensing, 2014, 94: 102-113.

Feng L, Jia Z, Zhang J. The dynamic monitoring of corn planting area distribution in response to climate change from 2001 to 2010: a case study of Northeast China[J]. Geografisk Tidsskrift-Danish Journal of Geography, 2016, 116(1): 44-55.

Feng L, Jia Z, Li Q. The dynamic monitoring of aeolian desertification land distribution and its response to climate change in northern China[J]. Scientific reports, 2016, 6(1): 1-10.

Sugihara G, May R, Ye H, et al. Detecting causality in complex ecosystems[J]. science, 2012, 338(6106): 496-500.

Q2: Fire identification requires high data accuracy. In this study, the impact of fire on regional GPP was explored, and whether the impact of multi-source data on fire range identification and GPP impact factor identification process was considered.

A2: Thank you very much for your good suggestion. Indeed, there is a certain uncertainty on fire range identification because of the impact of multi-source data. But the identification accuracy of this study can be assured. The validation of the NDFI is shown in Figure 4. Meanwhile, we have compared with the existing works during our study (Li et al., 2017), By comparing the existing works, we found the proposed NDFI is effective in burned area monitoring. According to your advices, we have added the related discussions to the revised manuscript, please see the Discussion and Conclusion part.

Li et al., 2017

Li M Z, Kang X R, Fan W Y. Burned area extraction in Huzhong forests based on remote sensing and the spatial analysis of the burned severity[J]. Scientia Silvae Sinicae, 2017, 53(3): 163-174.

Q3: From the result, the possibility of fire in Northeast China is high. Why? Did the author consider the reason?

A3: Thanks very much for your question. First, we have learned about the fire distribution in another article. Our result is consistent with the latest published article by Pang (Pang et al., 2022). In addition, longitude and latitude had the greatest influence on the occurrence of forest fires. This result is due to the uneven distribution of forest resources and regional differences in forest resources in China. The higher the vegetation cover (tree resources are abundant in northeast China) the more likely they are to cause problems related to forest fires. we have added the related discussions about the reason to the revised manuscript. Please see the Discussion and Conclusion part.

Pang Y, Li Y, Feng Z, et al. Forest Fire Occurrence Prediction in China Based on Machine Learning Methods[J]. Remote Sensing, 2022, 14(21): 5546.

Q4: In the discussion part, although the author discussed the impact of fire on GPP, there are still some deficiencies. In fact, there are many factors affecting GPP, which can be divided into natural and human factors. Many scholars try to separate the contribution rate of natural factors and human factors to GPP, such as What drives the vegetation dynamics in the Hengduan Mountain region, southwest China: Climate change or human activity? DOI: 10.1016/j.ecolind.2019.106013. Interestingly, fire can be either a natural factor or an artificial factor. From this point of view, can we discuss it further?

A4: Thanks for your questions. Indeed, forest fires are generated by a variety of factors. We will add more discussion and reference to the revised manuscript. Please see the discussion part in the revised manuscript.

This manuscript is a resubmission of an earlier submission. The following is a list of the peer review reports and author responses from that submission.

Round 1

Reviewer 1 Report

The paper 'A burned area dominated by natural factors will not bring large 
declines in forest ecosystem carbon flux in China' proposes a Normalized Difference Fire Index to estimate fire burned area. The proposed method is reasonable and applicable. After reading this paper, I think it is well organized and written. However, there are several claims and conclusions not fully supported or validated. Some techniques are not well explained. I would like to rate it as 'Major revision'. Kindly modify the content of this paper and clearly convey the technical details in future versions.

(1) This paper addresses an important problem of the causes and impacts of longtime burned areas on carbon cycling. Overall, the article is well organized and its presentation is good. However, What is the innovation of this researchI think it should be more specific.

(2) In the METHODS section, it would be beneficial to explain the relationship between the proposed method in Section 'Burned Area Monitoring Methods' and previous approaches. What is improved in the proposed method?

(3) In the RESULTS section, why close correlations mean the sensitive to the forest fire?

(4) In the DISCUSSIONS section, the authors should explain why they obtain these results. I mean the authors should compare with the existing works.

Reviewer 2 Report

Dear Authors,

Congratulations for your interesting research paper, which is very well written, demonstrating attention to details.

In addition, please explain what in TM? Is it only the band name or "Thematic Mapping" sensor? Same for MODIS and AVHRR, to be easily understood even by the non specialist readers.

Also check again the entire article, some phrases are not so clear, for example "The resolution is month, 0.05 degree, and the data unit is gcm-2. The nighttime-Light data were produced by a convolutional Long Short-Term Memory network".

References are written differently - with a huge space between the lines and without Italic for journal's name and volume. 

Please add "Conflicts of Interest" section,  as required by the journal.

Thank you for considering my opinion and good luck in your future activity!

The Reviewer

Reviewer 3 Report

The manuscript focused on the topics of burned area recognition, causes of the forest fire, effects of the forest fire on ecosystem carbon cycle. This would be a very meaningful work, if these problems were explained clearly. However, it is not perfect.

 (1)The title is inconsistent with the research content.

(2)The research area needs to be introduced.

(3)The huge forest fire that originated in China's Daxinganling Mountains in 1987 is not suitable to validate the validity of the NDFI. This can only indicate that NDFI will change in fire area, but it cannot indicate that NDFI will not change in non fire area.

(4)The relationship between NDFI and burned area needs to be explained in the method. How to recognize burned area by NDFI?

(5)What are the values, units and meanings listed in Table 1, 2 and 3? Why list these three months.

(6)Figure 2 needs to be explained.

(7)CCM method needs to be explained.

(8)The results are difficult to support the discussion and conclusions.

Reviewer 4 Report

This study used Convergent Cross Mapping (CCM) algorithm to analyze the causes of the forest fire and used Trend analysis to explore the impacts of forest fire on carbon flux. In general, the structure is reasonable and has certain application value. Some suggestions:

1.     The goal of this article is to find out the influencing factors of carbon flux. However, most of the contents of this article are about Forest Fire. This is inappropriate.

2.     Is it credible to use only the range of one fire as the correct verification basis for extracting the range of possible forest fires across the country?

3.     Ecosystem Carbon Cycle is equal to Ecosystem Carbon flux? The statements in the article should be consistent.

4.     Effects of the forest Fire on Ecosystem Carbon Cycle: This part is too thin to dig out more effective information behind the research results, such as spatial heterogeneity differences, influence factor differences, etc.

5.     Language needs polishing.